# Neonatal Pyruvate Kinase Deficiency Presenting with Severe Hemolytic Anemia and Liver Failure

**DOI:** 10.3390/children12111539

**Published:** 2025-11-14

**Authors:** Yung-Han Hsu, Chuen-Bin Jiang, Jen-Yin Hou, Wai-Tim Jim, Shuan-Pei Lin, Szu-Wen Chang, Kai-Ti Tseng, Ni-Chung Lee

**Affiliations:** 1Department of Pediatrics, Hsinchu Municipal MacKay Children’s Hospital, Hsinchu City 300046, Taiwan; 2Department of Pediatric Gastroenterology, Hepatology and Nutrition, MacKay Children’s Hospital, Taipei City 104217, Taiwan; 3Department of Medicine, MacKay Medical College, MacKay Medical University, New Taipei City 252005, Taiwan; 4Division of Pediatric Hematology-Oncology, MacKay Children’s Hospital, Taipei City 104217, Taiwan; 5Division of Neonatology, Department of Pediatrics, MacKay Children’s Hospital, Taipei City 104217, Taiwan; 6Department of Pediatric Genetics, MacKay Children’s Hospital, Taipei City 104217, Taiwan; 7Division of Neonatology, Department of Pediatrics, MacKay Memorial Hospital Tamshui Branch, New Taipei City 251404, Taiwan; 8Department of Medical Genetics, National Taiwan University Hospital, Taipei City 100225, Taiwan

**Keywords:** pyruvate kinase deficiency, *PKLR* mutation, neonatal liver failure, hemolytic anemia, conjugated hyperbilirubinemia, genetic diagnosis, iron overload, hepatic involvement

## Abstract

**Background****:** Pyruvate kinase deficiency (PKD) is the most prevalent enzymatic defect of the glycolytic pathway, causing chronic congenital non-spherocytic hemolytic anemia. Clinical severity ranges from mild anemia to transfusion-dependent diseases. Severe neonatal presentations, including liver failure, have rarely been reported. **Case presentation:** We report a preterm female neonate with PKD who developed early-onset hemolytic anemia, conjugated hyperbilirubinemia, hepatosplenomegaly, coagulopathy, and progressive transaminitis. Imaging demonstrated hepatomegaly with diffuse parenchymal involvement. Whole-genome sequencing identified compound heterozygous pathogenic mutations in the *PKLR* gene, confirming the diagnosis of PKD. The patient required continuous transfusion support and was discharged following clinical stabilization. **Discussion:** Although PKD most often manifests as isolated hemolytic anemia, this case illustrates a rare neonatal phenotype with concurrent liver dysfunction. We investigated the potential underlying mechanism. Recognition of hepatic involvement in PKD is essential because liver failure is associated with considerable morbidity and mortality, and it may necessitate interventions such as liver transplantation. **Conclusions:** This case highlights the importance of considering PKD in neonates presenting with hemolysis and liver failure. Early genetic confirmation enables timely management, including transfusion support, iron overload surveillance, and anticipatory guidance for potential hepatic complications.

## 1. Introduction

Pyruvate kinase (PK) deficiency is an autosomal recessive disorder and the most common enzyme defect of the glycolytic pathway [1,2]. It is the leading cause of chronic congenital non-spherocytic hemolytic anemia globally, with an estimated prevalence of approximately 1 in 100,000 to 1 in 300,000 individuals [3]. The clinical spectrum of pyruvate kinase deficiency (PKD) is highly heterogeneous, ranging from mild, manageable anemia to severe transfusion-dependent disease or fatal hemolytic anemia [4]. Only a few cases of severe neonatal PK deficiency with liver failure have been reported [5]. To address this gap, we describe a rare neonatal presentation of PKD with severe hemolytic anemia and liver failure.

## 2. Case Presentation

A 1-day-old preterm female neonate (gestational age [GA] 35 + 2 weeks) was admitted with respiratory distress. The birth weight of the neonate was 2255 g, and the Apgar score was 9 at the first minute and 10 at the fifth minute. Prenatal examination showed no abnormality, and there was no aberrant maternal pregnancy history. Laboratory evaluation after birth revealed macrocytic anemia with a hemoglobin level of 8.5 g/dL, mean corpuscular volume (MCV) of 137 fL, and reticulocyte count of 42.3%. There was no thrombocytopenia (platelet count 173,000/µL). At 10 h after birth, hyperbilirubinemia was detected with a total bilirubin level of 15.6 mg/dL and a direct bilirubin level of 0.8 mg/dL. Liver function tests showed markedly elevated aspartate aminotransferase (AST) and alanine aminotransferase (ALT) levels of 1462 and 333 IU/L, respectively. On physical examination, the neonate was found to have icteric sclera, hepatomegaly extending two fingerbreadths below the right costal margin, and splenomegaly extending 1.5 fingerbreadths below the left costal margin.

Hematuria was detected 17 h after birth, accompanied by proteinuria (2+), bilirubinuria (1+), occult blood (3+), and 1 red blood cell per high-power field (RBC/HPF) on urinalysis. Blood tests demonstrated coagulopathy with a prothrombin time (PT) of 21.9 s and an international normalized ratio (INR) of 2.17, and therefore, vitamin K was administered. Despite the treatment, the INR remained elevated (2.28) the following day. At 24 h post-birth, the AST/ALT levels surged (4139/506 IU/L), and by day 3, conjugated hyperbilirubinemia had developed, with a total bilirubin level of 16.9 mg/dL and direct bilirubin level of 3.7 mg/dL. Therefore, treatment with silymarin, ursodeoxycholic acid, and phenobarbital was initiated.

Hemolytic anemia was suspected owing to markedly elevated lactate dehydrogenase (LDH) level (>12,000 IU/L). Further investigations included abdominal sonography, which demonstrated hepatomegaly with heterogeneous echogenicity of the liver with more significant hypoechoic density of the left lobe. To further investigate the lesions identified on sonography, an abdominal CT scan was performed, which demonstrated hepatomegaly with heterogeneous echogenicity, measuring approximately 6.7 cm in length on the coronal plane and revealed infiltrative hypodense lesions larger than 6.6 cm in diameter involving multiple liver segments (Figure 1). Subsequently, an abdominal MRI with Primovist contrast was conducted to further evaluate the hepatic tumor suspected based on the CT scan, which showed ill-defined fatty signals affecting the entire liver (Figure 2), leading to a differential diagnosis that included steatosis and hemochromatosis.

Both direct and indirect antiglobulin tests were negative, and glucose-6-phosphate dehydrogenase (G6PD) deficiency was excluded. Peripheral blood smear demonstrated numerous normoblasts with marked anisocytosis and poikilocytosis (Figure 3).

Comprehensive infection screening, including toxoplasmosis, syphilis, rubella, cytomegalovirus, and herpes simplex (TORCH), group B *Streptococcus* (GBS), influenza, enterovirus, parvovirus, and viral hepatitis, as well as urine viral culture, yielded negative results. To explore the cause of neonatal liver failure, we measured serum ferritin level, transferrin saturation, and transferrin level to rule out neonatal hemochromatosis, showing markedly elevated transferrin saturation (96.34%) and ferritin level (248.88 ng/mL). Owing to high suspicion for neonatal hemochromatosis or an underlying metabolic disorder, a molecular work-up was performed. This led to the discovery of two pathogenic *PKLR* gene mutations (c.1015dupG [p.Asp339GlyfsTer62] and c.694G>C [P.Gly232Arg]) through whole-exome sequencing (Table 1). The HFE gene test was negative, and therefore, hemochromatosis was ruled out. The female infant was discharged after 2 months of hospitalization with supportive care. At 3 months of age, she developed sepsis, with blood cultures positive for methicillin-resistant *Staphylococcus aureus* (MRSA). During this episode, her hemoglobin level dropped to 4.7 g/dL, and she exhibited severe cholestasis with a total bilirubin level of 45.8 mg/dL and a direct bilirubin level of 28.8 mg/dL. Following antibiotic therapy and supportive care, the bilirubin levels declined. The patient currently requires monthly blood transfusions to maintain hemoglobin levels above 8.0 g/dL after discharge. Monthly monitoring of serum ferritin showed levels ranging from 781 to 1672 ng/mL. Iron chelation with deferoxamine was initiated during RBC transfusions at 18 months of age, and oral deferasirox was started at 2 years of age. Liver function gradually improved, and by 1 year of age, the total bilirubin level had decreased to 1.5 mg/dL, the direct bilirubin level to 0.4 mg/dL, the AST level to 32 IU/L, and the ALT level to 24 IU/L, allowing for discontinuation of ursodeoxycholic acid, phenobarbital, and silymarin. The trajectory of laboratory test results from birth to 3 years of age is shown in Table 2. Close monitoring of complications related to chronic transfusion will be necessary in the future. Furthermore, liver transplantation for PKD is considered only after conventional treatments, such as transfusion therapy and splenectomy, have failed. This procedure is reserved for severe cases involving refractory hemolytic anemia, transfusion-dependent iron overload, or serious liver complications that persist despite splenectomy. It is a rare, last-resort option owing to the risks associated with surgery and the need for lifelong immunosuppression.

## 3. Discussion

Hemolysis refers to the pathological shortening of red blood cell lifespan, in contrast to the normal physiological removal of aging erythrocytes through senescence [6]. Unconjugated hyperbilirubinemia is a common complication of hemolytic anemia. Under normal conditions, bilirubin circulates in the bloodstream bound to albumin; however, in neonates with hereditary hemolytic anemia (HHA), a rapid rise in unconjugated bilirubin level may exceed the binding capacity of albumin, allowing free bilirubin to cross the blood–brain barrier and increase the risk of neurotoxicity [2]. In the present case, both direct and indirect antiglobulin tests were negative. The coexistence of jaundice and signs of hemolysis in the neonate strongly suggests an intrinsic erythrocyte defect [7].

Glycolysis is the primary pathway for ATP generation in healthy red blood cells [8]. Pyruvate kinase plays a crucial role in this process by catalyzing the conversion of phosphoenolpyruvate to pyruvate, thereby producing ATP [2,5,8,9]. Adequate ATP synthesis is essential for maintaining the structural stability and functional competence of erythrocytes [8]. Insufficient or abnormal pyruvate kinase activity results in reduced ATP production, leading to impaired membrane deformability, cellular dehydration, and premature erythrocyte destruction in the spleen or liver [8]. PKD represents the leading cause of hereditary non-spherocytic hemolytic anemia [10]. PKD results from autosomal recessive variants in the *PKLR* gene, located on chromosome 1q21 [3,8].

Evaluation for PKD is recommended in all patients with non-immune hemolytic anemia once hemoglobinopathies and erythrocyte membrane disorders have been excluded [3]. Laboratory diagnosis of PKD relies on demonstrating reduced pyruvate kinase activity and identifying pathogenic mutations in the *PKLR* gene [3]. PK activity in red blood cell lysates can be measured using a spectrophotometric assay [7]. However, PK activity does not correlate with disease severity; therefore, DNA analysis of the *PKLR* gene is recommended to confirm the diagnosis [7].

Unlike the typical presentation of hemolysis in PKD, which is associated with unconjugated hyperbilirubinemia, our patient (infant) developed conjugated hyperbilirubinemia 3 days after birth, accompanied by markedly elevated AST and ALT levels. In addition, liver failure was confirmed by a persistently elevated INR despite vitamin K administration.

PK activity in humans is carried out by four isoenzymes encoded by two genes [9,11]. The *PKLR* gene on chromosome 1q21 encodes both PK-L (expressed in the liver) and PK-R (present in mature red blood cells) [2,4,5,9,11]. In contrast, the *PK m* gene, located on chromosome 15, encodes two isoenzymes: PK-M1 (found in skeletal muscles, heart, and brain) and PK-M2 (expressed in proliferating fetal tissues and red blood cell precursors) [9,11]. PK-M2 is also expressed in immature hepatocytes; however, as cells differentiate, PK-M2 is replaced by PK-R and PK-L [11]. Mutations in the *PKLR* gene affect both erythrocyte and liver isoforms; however, clinical manifestations are generally restricted to red blood cells, likely due to retained protein synthesis capacity and residual PK-M2 activity in hepatocytes [2,8]. Typically, PKD does not cause liver dysfunction because the reduction in enzymatic activity is compensated by ongoing enzyme synthesis in hepatocytes [5]. In contrast, mature red blood cells rely exclusively on glycolysis and are dependent on PK-R [8]. In a small subset of individuals with severe PKD, persistent expression of PK-M2 isoenzyme has been observed in both mature red blood cells and the liver. This phenomenon may act as a compensatory mechanism to reduce hemolysis and prevent liver dysfunction [11].

Liver failure is an uncommon complication in neonates with PKD, and several hypotheses have been proposed to explain the liver failure and cholestasis observed in this condition [8]. First, a combined deficiency of the PK-L and PK-M2 isozymes may lead to ATP depletion in hepatocytes, presenting as transaminitis and conjugated hyperbilirubinemia. This may further progress to hepatocellular injury and impaired synthetic function [8]. Although the mortality rate of patients with this manifestation is high, successful management through liver transplantation has been reported [8]. Second, intense hemolysis could impair the excretory function of an initially healthy liver, potentially causing reflux of the conjugated bilirubin [12]. Antenatal hemolysis, combined with bile obstruction of the intrahepatic bile ducts, may lead to postnatal ductal paucity [5]. Persistent obstruction of the canalicular and small ducts, combined with intrahepatic cholestasis, contributes to progressive hyperbilirubinemia and liver disease [5]. Third, extramedullary hematopoiesis in the liver, accompanied by infiltration of extrahepatic monocytes and secretion of cytokines, can induce inflammation and compromise the integrity of the intrahepatic canaliculi [11]. Finally, iron toxicity may contribute to liver failure, as iron overload resulting from chronic hemolysis, repeated transfusions, and impaired erythropoiesis increases intestinal iron uptake, although excessive accumulation is generally not observed in the neonatal period [11]. PKD can lead to iron overload even without transfusions, suggesting a potential link between the accumulation of iron and abnormalities in hepcidin, a key regulator of iron homeostasis [11,13]. A recent retrospective analysis of PKD-related complications demonstrated that the incidence of liver cirrhosis did not significantly differ according to the patient’s transfusion history [14]. Liver iron overload was identified in 62% of patients with PKD, independent of transfusion history [13,14]. Screening for iron overload using serum ferritin is recommended in both pediatric and adult patients with PKD, irrespective of transfusion history, to facilitate early detection and prevention of complications [3].

The occurrence of liver failure in our case is likely due to multiple factors, and several possible mechanisms can be considered. The most plausible explanations include severe hemolysis accompanied by cholestasis, as well as bile duct injury and inflammation resulting from extramedullary hematopoiesis, especially because there is no evidence of a PK-M2 gene variant. Iron toxicity is considered less likely, as it is typically not observed during the neonatal period.

While reviewing the literature, we found that only a few neonates with PKD have developed hepatic failure. Lin et al. reported a term neonate with PKD who had persistent pulmonary hypertension [15]. Laboratory findings demonstrated hemolytic anemia, thrombocytopenia, and conjugated hyperbilirubinemia, but no evidence of hepatic failure. Similarly, Dulmovits et al. described a male neonate with PKD who presented with hemolytic anemia, thrombocytopenia, and cholestasis, but without hepatic failure [16]. Hou et al. reported the first Taiwanese male neonate diagnosed with PKD, who did not exhibit either cholestasis or hepatic failure [4]. In contrast, Olivier et al. described a male neonate with PKD whose clinical presentation closely resembled that of our case, including severe hemolytic anemia (Hb 4.8 g/dL; LDH 5411 U/L), thrombocytopenia (platelets 41 × 10^3^/µL), cholestasis, and hepatic failure [17]. The diagnosis was confirmed by both reduced PK enzyme activity (0.675 U/g Hb) and identification of *PKLR* mutations. Unfortunately, the infant died of sepsis at 3 months of age. Unlike most of these previously reported cases, our female patient did not develop thrombocytopenia. The clinical information of the previously reported neonatal PKD cases is summarized in Table 3.

This report has certain limitations. It describes a single case and does not provide long-term outcome data.

## 4. Conclusions

PKD is a chronic hemolytic anemia with diverse clinical manifestations. Diagnosis is generally established through enzyme activity assays and molecular genetic analysis. Liver failure and cholestasis in newborns with PKD are rare but may arise from hepatocellular ATP depletion, severe hemolysis with cholestasis and bile duct damage, inflammation due to extramedullary hematopoiesis, or early onset iron toxicity. It is important to consider PKD in neonates presenting with hemolysis and liver failure. Early genetic confirmation facilitates timely management, encompassing transfusion support, monitoring for iron overload, and proactive guidance regarding potential hepatic complications.

## Figures and Tables

**Figure 1 children-12-01539-f001:**
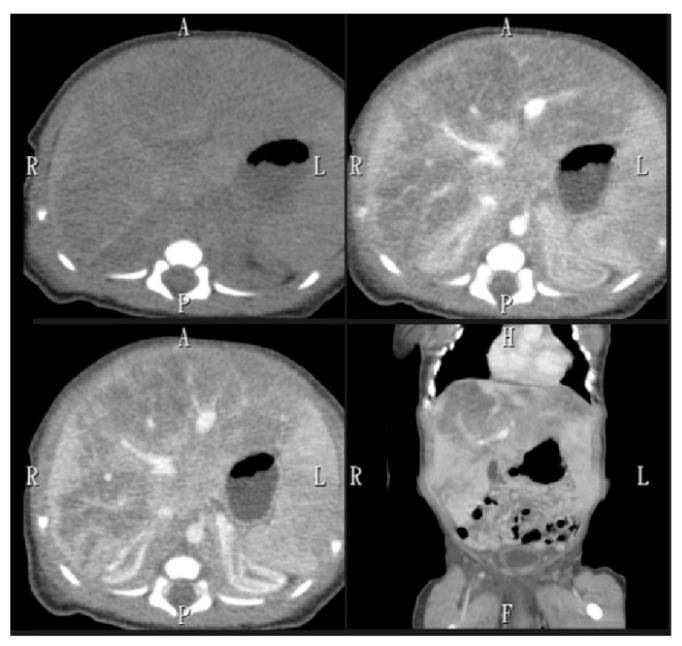
Abdominal CT demonstrated hepatomegaly and infiltrative hypodense lesions in the liver, visible both with and without contrast.

**Figure 2 children-12-01539-f002:**
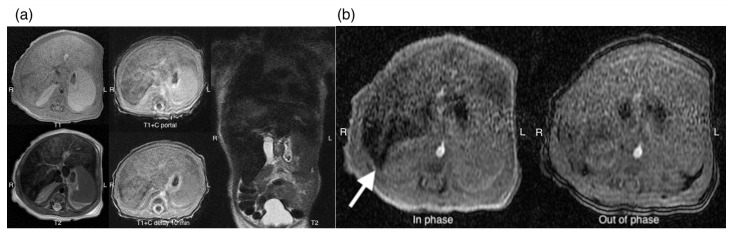
(**a**) Abdominal MRI with Primovist in T1- and T2-weighted sequences demonstrated suspicious diffuse fatty infiltration of the liver, with relative sparing of segment 5 (S5); (**b**) in-phase imaging demonstrated a signal drop in the hepatic angle (arrow).

**Figure 3 children-12-01539-f003:**
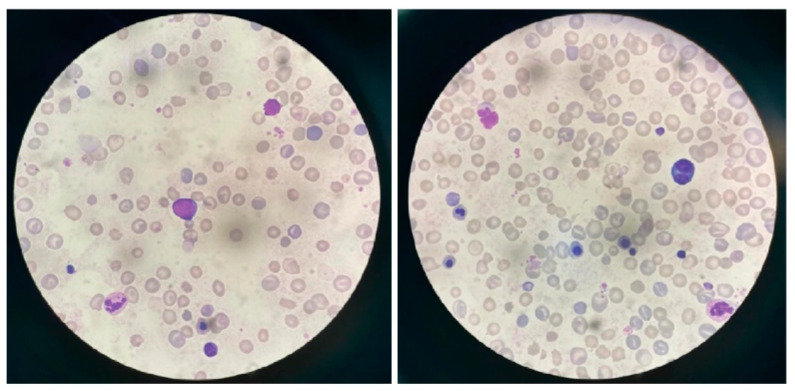
A blood smear demonstrated numerous normoblasts with marked severe anisocytosis and poikilocytosis.

**Table 1 children-12-01539-t001:** Genetic variants detected in the patient using whole-exome sequencing.

Gene	Physical Position	Transcript	Exon	Nucleotide (AA Change)	Zygosity	MAF	CADD Phred Score	ClinVar	ACMG Interpretation	Inheritance
PKLR	1:155,294,335	NM_000298.6	Exon7	c.1015dupG (p.Asp339GlyfsTer62)	Het	8.96 × 10^−6^	NA	NA	Pathogenic	Mother
PKLR	1:155,295,116	NM_000298.6	Exon5	c.694G>C (p.Gly232Arg)	Het	NA	35	Likely Pathogenic	Likely Pathogenic	Father

NA: not available; MAF: maximum minor allele frequency.

**Table 2 children-12-01539-t002:** Trajectory of laboratory test results from birth to 3 years of age.

	birth	10 h/o	24 h/o	48 h/o	72 h/o	7 d/o	1 m/o	3 m/o	1 y/o	2 y/o	3 y/o
Hb (g/dL)	8.5		11.6	10.3	9.2	9.9	8.2	4.7	8.4	9.3	8.0
Platelet (/µL)	173,000		89,000	87,000	79,000	90,000	278,000	219,000	306,000	309,000	395,000
AST (IU/L)		1462	4139	3001	1040	30	93	929	32	42	47
ALT (IU/L)		333	506	361	283	54	69	401	24	59	32
Total bilirubin (mg/dL)		15.6	16.6	16.9	13.5	12.2	11.1	45.8	1.5	1.8	2.4
Direct bilirubin (mg/dL)		0.8	1.2	3.7	5.8	7.5	6.4	28.8	0.4	0.4	0.5

**Table 3 children-12-01539-t003:** Clinical information of previously reported neonatal PKD cases.

	GA (Weeks)	BBW	Clinical Manifestations	Associated Diagnosis	Thrombo-cytopenia	Hepatic Failure	Cholestasis	Diagnosis Method	Treatment	Outcome
Lin et al. [15]	37 + 6	2.4 kg	Severe dyspnea, extreme anemia, skin pallor, jaundice, and hypoxemia	Heart failure, persistent pulmonary hypertension in the neonate (PPHN)	Yes	No	Yes	Whole-exome sequencing	Mechanical ventilator, RBC transfusion, nutrition support	No significant complications
Dulmovits et al. [16]	Term	3.6 kg	Respiratory distress, abdominal distension, jaundice, hepatosplenomegaly, petechiae, anemia, and a diffuse blue macular rash	Incomplete neonatal Kawasaki disease	Yes	No	Yes	Whole-exome sequencing	Platelet transfusions, intensive phototherapy, double volume exchange transfusion, aspirin, IVIG	Stabilized, transfusion- dependent
Hou et al. [4]	NA	NA	Respiratory distress and anemia	NA	No	No	No	Whole-exome sequencing	RBC transfusion	Stabilized, transfusion- dependent
Olivier et al. [17]	Term	3.4 kg	Hypotonic, generalized edema, and anemia	NA	Yes	Yes	Yes	PK enzyme level, genetic study	Blood transfusion, exchange transfusion	Died of sepsis at 3 months old

GA: gestational age; BBW: birth body weight; NA: not available.

## Data Availability

Data are contained within the article.

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
