# Peer review of "Neonatal Pyruvate Kinase Deficiency Presenting with Severe Hemolytic Anemia and Liver Failure"

_children, 2025, doi:10.3390/children12111539_

Round 1

Reviewer 1 Report

Comments and Suggestions for Authors

1.Page 2 Line 70: Is there any reference or guideline to start this medications for liver failure? If not please clear the main reason for giving these to neonates. “Therefore, treatment with silymarin, ursodeoxycholic 70 acid, and phenobarbital was initiated”

2. If an hepatomegaly diagnosed why these two additional tests were made? “Abdominal computed tomography (CT) confirmed hepatomegaly with heterogeneous echogenicity, measuring approximately 6.7 cm in length on the coronal plane and revealed infiltrative hypodense lesions larger than 6.6 cm in diameter involving multiple liver segments. Furthermore, an abdominal magnetic resonance imaging (MRI) showed ill defined fatty signals affecting the entire liver.”

3. Both direct and indirect antiglobulin tests were negative, and glucose-6-phosphate 88 dehydrogenase (G6PD) was excluded OR “G6PD deficiency was excluded?”

4. When a clinican should get genetic analysis with anemia and hemolysis? After hematology colsuntation? After all enzyme assays? After Ultrasonograhy? Authors should get a diagram to show the list and managment form a guideline?

5. What about a hospital without genetic screening tests? When they should send the baby with anemia and hemolysis? Or else?

Author Response

[Question 1]: Page 2 Line 70: Is there any reference or guideline to start this medications for liver failure? If not please clear the main reason for giving these to neonates. “Therefore, treatment with silymarin, ursodeoxycholic acid, and phenobarbital was initiated”

[Response 1]:

A narrative review published in Advances in Therapy (2020) reported that silymarin can reduce oxidative stress and consequent cytotoxicity, thereby protecting intact hepatocytes or hepatocytes not yet irreversibly damaged. The mechanisms underlying the beneficial effects of ursodeoxycholic acid (UDCA) in cholestatic disorders are being increasingly elucidated. Paumgartner et al. (Hepatology, 2002) proposed three major mechanisms of action based on experimental evidence: (1) protection of cholangiocytes from the cytotoxic effects of hydrophobic bile acids, (2) stimulation of hepatobiliary secretion, and (3) protection of hepatocytes from bile acid-induced apoptosis. One or more of these mechanisms may play a role in different cholestatic disorders and at various stages of disease progression. Zhang et al. (BMC Pediatrics, 2025) reported that phenobarbitone is both safe and effective in lowering serum bilirubin levels in neonates. Phenobarbitone alleviates jaundice by promoting bilirubin excretion, enhancing glucuronidation through the induction of hepatic microsomal enzymes, and increasing the production of receptor proteins involved in bilirubin uptake. Both phenobarbitone and phototherapy are effective in reducing the peak serum bilirubin (PSB) level. These references support our rationale for initiating treatment with the aforementioned medications in this case. However, we did not include detailed descriptions of these drugs in the manuscript, as the primary purpose of the article is to report a case of neonatal pyruvate kinase deficiency presenting with severe liver failure and to discuss the potential underlying mechanisms. Elaborating on the pharmacologic mechanisms of these medications would be beyond the scope of the case report and potentially redundant.

[Question 2]: If an hepatomegaly diagnosed why these two additional tests were made? “Abdominal computed tomography (CT) confirmed hepatomegaly with heterogeneous echogenicity, measuring approximately 6.7 cm in length on the coronal plane and revealed infiltrative hypodense lesions larger than 6.6 cm in diameter involving multiple liver segments. Furthermore, an abdominal magnetic resonance imaging (MRI) showed ill defined fatty signals affecting the entire liver.”

[Response 2]:

We appreciate the insightful comment. Abdominal ultrasonography revealed hepatomegaly with heterogeneous echogenicity of the liver with more significant hypoechoic density of the left lobe. To further investigate the lesions identified on sonography, an abdominal CT scan was performed, which demonstrated hepatomegaly with infiltrative hypodense lesions in the liver. Subsequently, an abdominal MRI with Primovist contrast was conducted to further evaluate the hepatic tumor identified on CT (Page 2, Paragraph 4, Line 80-82, 85-89).

[Question 3]: Both direct and indirect antiglobulin tests were negative, and glucose-6-phosphate dehydrogenase (G6PD) was excluded OR “G6PD deficiency was excluded?”

[Response 3]:

As you pointed out, glucose-6-phosphate dehydrogenase (G6PD) deficiency was excluded. We apologize for the mistake. We have corrected the mistake in the manuscript (Page 3, Paragraph 1, Line 99).

[Question 4]: When a clinican should get genetic analysis with anemia and hemolysis? After hematology colsuntation? After all enzyme assays? After Ultrasonograhy? Authors should get a diagram to show the list and managment form a guideline?

[Response 4]:

Thank you for the excellent comment. In this case, genetic analysis was considered for neonatal cholestasis after excluding obstructive and infectious causes, given the suspicion of an inherited cholestatic or metabolic disorder. Specifically, genetic testing for hereditary hemochromatosis was performed owing to the markedly elevated transferrin saturation (96.34%) and ferritin level (248.88 ng/mL), as well as the possible metabolic etiology of liver failure. The HFE gene test result was negative, thereby excluding this diagnosis (Page 4, Paragraph 1, Lines 108-111, 114-115).

[Question 5]. What about a hospital without genetic screening tests? When they should send the baby with anemia and hemolysis? Or else?

[Response 5]:

This situation is similar to our experience in the present case. Once a metabolic cause of liver failure was suspected, blood was sampled and sent to a medical center equipped for genetic analysis.

We hope that the revised manuscript is suitable for publication in Children.

Reviewer 2 Report

Comments and Suggestions for Authors

This case report describes a preterm neonate with pyruvate kinase deficiency (PKD) presenting with severe haemolytic anaemia and liver failure, a rare neonatal phenotype. The manuscript is well-structured and clinically relevant but requires improvements in scientific rigor, data presentation, and discussion depth.

Several critical clinical parameters are missing or inadequately presented:

  1. Lab tests: Platelet counts not reported (only mentioned that thrombocytopenia was absent), incomplete timeline of bilirubin evolution. Evaluate adding a comprehensive Table 2 with complete laboratory evolution (CBC, liver enzymes, coagulation, bilirubin) at key timepoints: birth, 24h, 48h, 72h, 1 week, 1 month, discharge, and follow-up visits, if available. What is the current developmental status of the child at 2 years?
  • Personal history: Birth weight, Apgar scores, and maternal pregnancy history are absent. Is any data available?
  • For the genetic test: The c.694G>C mutation location is inconsistent (the abstract says c.694G>C, table says c.94G>C). Please clarify. No parental confirmation of inheritance pattern was provided, and there was no discussion of genotype-phenotype correlation for these specific variants or in silico predictions. Consider adding these points to complete the discussion of the case report, and discussing whether these mutations have been previously reported and their known phenotypes
  • PK enzyme activity was never measured (only mentioned as a diagnostic method). Explain why PK enzyme activity was not measured (unavailable? Already proceeded to genetic testing?) and include iron parameters at presentation (ferritin, transferrin saturation, serum iron), if available.
  • Figures 1 and 2 show significant hepatic changes but lack quantitative assessment, and "Fatty infiltration" is mentioned but not confirmed by biopsy. Add arrows/annotations to images highlighting key findings, liver and spleen volume measurements and follow-up images, if available. If fatty infiltration is suspected, explain why biopsy or MRI fat fraction quantification was not performed
  • The discussion presents multiple hypotheses for liver failure, but doesn't integrate them; there is no discussion of why THIS patient developed liver failure while most PKD patients do not. Consider adding a dedicated section discussing "Why liver failure in this case?" considering specific mutations and their functional consequences, prematurity and hepatic immaturity, potential PK-M2 expression deficiency (if data available)
  • Literature Review: Only 4 similar cases are discussed (lines 192-204), no systematic review of all reported PKD cases with liver failure. Consider creating a Table to discuss these cases.
  • Treatment and Management: transfusion regimen not specified (volume, frequency, targets); chelation therapy started late (18 months for deferoxamine), so justify chelation timing based on ferritin trends. Why was liver transplant not considered? Discuss transplant consideration/decision-making and add a brief section on emerging therapies and potential future applications

Author Response

[Question 1]:

Lab tests: Platelet counts not reported (only mentioned that thrombocytopenia was absent), incomplete timeline of bilirubin evolution. Evaluate adding a comprehensive Table 2 with complete laboratory evolution (CBC, liver enzymes, coagulation, bilirubin) at key timepoints: birth, 24h, 48h, 72h, 1 week, 1 month, discharge, and follow-up visits, if available. What is the current developmental status of the child at 2 years?

[Response 1]:

We appreciate the insightful comment. The platelet count was 173000/µL at birth, and the trajectory of laboratory test results from birth to 3 years of age is shown in the table below. We have included the table in the manuscript (Page 2, Paragraph 2, Lines 61-62; Page 4, Paragraph 1, Lines 128-129; Page 4-5 Table 2.)

Birth

10 h/o

24 h/o

48 h/o

72 h/o

7 d/o

1 m/o

3 m/o

1 y/o

2 y/o

3 y/o

Hb (g/dL)

8.5

11.6

10.3

9.2

9.9

8.2

4.7

8.4

9.3

8.0

platelet (/µL)

173000

89000

87000

79000

90000

278000

219000

306000

309000

395000

AST (IU/L)

1462

4139

3001

1040

30

93

929

32

42

47

ALT (IU/L)

333

506

361

283

54

69

401

24

59

32

Total bilirubin (mg/dL)

15.6

16.6

16.9

13.5

12.2

11.1

45.8

1.5

1.8

2.4

Direct bilirubin (mg/dL)

0.8

1.2

3.7

5.8

7.5

6.4

28.8

0.4

0.4

0.5

[Question 2]: Personal history: Birth weight, Apgar scores, and maternal pregnancy history are absent. Is any data available?

[Response 2]:

Thank you for the thoughtful comment. The birth weight of the baby was 2255 g and Apgar score was 9 at the first minute and 10 at the fifth minute. Her mother had no history of GDM, PIH, preeclampsia, APH, and PPH, and the family denied history of thalassemia, G6PD deficiency, and thyroid disease. Antepartum screening tests for HIV, hepatitis B virus (HBsAg and HBeAg), and syphilis were negative. Vaginal GBS culture was not performed. Results of routine fetal ultrasonography were normal. Level II ultrasonography and amniocentesis were not conducted. The mother denied social history of smoking habit, alcohol consumption, or drug abuse. We have added the history in the manuscript (Page 2, Paragraph 2, Lines 57–59).

[Question 3]: For the genetic test: The c.694G>C mutation location is inconsistent (the abstract says c.694G>C, table says c.94G>C). Please clarify. No parental confirmation of inheritance pattern was provided, and there was no discussion of genotype-phenotype correlation for these specific variants or in silico predictions. Consider adding these points to complete the discussion of the case report, and discussing whether these mutations have been previously reported and their known phenotypes

[Response 3]:

Thanks for the excellent and valuable comment. The pathogenic PKLR gene mutation is c.694G>C, rather than c.94G>C. We have made the necessary revision in the table (Page 4, Table 1.) Furthermore, our literature review did not reveal any reports that described these specific variants.

[Question 4]: PK enzyme activity was never measured (only mentioned as a diagnostic method). Explain why PK enzyme activity was not measured (unavailable? Already proceeded to genetic testing?) and include iron parameters at presentation (ferritin, transferrin saturation, serum iron), if available.

[Response 4]:

Thanks for the excellent comment. The measurement of PK enzyme activity is not performed in our hospital or the nearby medical center. To explore the cause of neonatal liver failure, we measured serum ferritin level, transferrin saturation, and transferrin level to rule out neonatal hemochromatosis. Given the markedly elevated transferrin saturation (96.34%) and ferritin level (248.88 ng/mL), hereditary hemochromatosis was suspected, prompting consideration of genetic analysis along with evaluation for metabolic causes of liver failure. The HFE gene test result was negative, effectively ruling out hereditary hemochromatosis (Page 4, Paragraph 1, Lines 108-111, 114-115).

[Question 5]: Figures 1 and 2 show significant hepatic changes but lack quantitative assessment, and "Fatty infiltration" is mentioned but not confirmed by biopsy. Add arrows/annotations to images highlighting key findings, liver and spleen volume measurements and follow-up images, if available. If fatty infiltration is suspected, explain why biopsy or MRI fat fraction quantification was not performed

[Response 5]:

Thank you for the thoughtful comment. We conducted an abdominal MRI with Primovist to further investigate a hepatic tumor suspected based on an abdominal CT scan, which indicated hepatomegaly and infiltrative hypodense lesions in the liver. The MRI showed ill-defined fatty signals affecting the entire liver, leading to a differential diagnosis that included steatosis and hemochromatosis (Page 2, Paragraph 4, Lines 80-82, 85-89). To explore the cause of neonatal liver failure, we measured serum ferritin level, transferrin saturation, and transferrin level to rule out neonatal hemochromatosis. Given the markedly elevated transferrin saturation (96.34%) and ferritin level (248.88 ng/mL), hereditary hemochromatosis was suspected, prompting consideration of genetic analysis along with evaluation for metabolic causes of liver failure. The HFE gene test result was negative, effectively ruling out hereditary hemochromatosis (Page 4, Paragraph 1, Lines 108-111, 114-115). A liver biopsy was initially considered to evaluate the cause of liver failure and the abnormal findings observed on MRI. However, after confirming the diagnosis of pyruvate kinase deficiency (PKD), it became apparent that the liver injury was likely due to severe hemolysis associated with PKD. Additionally, as cholestasis and liver function gradually improved with supportive care, the liver biopsy was ultimately deemed unnecessary and was not performed.

[Question 6]: The discussion presents multiple hypotheses for liver failure, but doesn't integrate them; there is no discussion of why THIS patient developed liver failure while most PKD patients do not. Consider adding a dedicated section discussing "Why liver failure in this case?" considering specific mutations and their functional consequences, prematurity and hepatic immaturity, potential PK-M2 expression deficiency (if data available)

[Response 6]:

Thanks for the excellent comment. The development of liver failure in our case is likely due to multiple factors, and several possible mechanisms have been considered. The most plausible explanations include severe hemolysis accompanied by cholestasis, as well as bile duct injury and inflammation resulting from extramedullary hematopoiesis, especially because there is no evidence of a PK-M2 gene variant. Iron toxicity is considered less likely, as it is typically not observed during the neonatal period (Page 6, Paragraph 3, Lines 214–219).

[Question 7]: Literature Review: Only 4 similar cases are discussed (lines 192-204), no systematic review of all reported PKD cases with liver failure. Consider creating a Table to discuss these cases.

[Response 7]:

We appreciate the insightful comment. We have organized the clinical information of previously reported neonatal PKD cases as follows. The table has been included in the manuscript (Page 7, Paragraph 1, Lines 233-235, Table 3).

GA (weeks)

BBw

Clinical manifestations

Associated diagnosis

Thrombo-cytopenia

Hepatic failure

Cholestasis

Diagnosis method

Treatment

Outcome

Lin et al.

37+6

2.4 kg

Severe dyspnea, extreme anemia, skin pallor, jaundice, and hypoxemia

Heart failure, persistent pulmonary hypertension in the neonate (PPHN)

Yes

No

Yes

Whole exome sequencing

Mechanical ventilator, RBC transfusion, nutrition support

No significant complications

Dulmovits et al.

Term

3.6 kg

Respiratory distress, abdominal distension,

jaundice, hepatosplenomegaly, petechiae, anemia, and a diffuse blue macular rash

Incomplete neonatal

Kawasaki disease

Yes

No

Yes

Whole exome sequencing

Platelet transfusions, intensive

phototherapy, double volume exchange transfusion, aspirin, IVIG

Stabilized, transfusion-

dependent

Hou et al.

NA

NA

Respiratory distress, and anemia

NA

No

No

No

Whole exome sequencing

RBC transfusion

Stabilized, transfusion-

dependent

Olivier et al.

Term

3.4 kg

Hypotonic, generalized edema, and anemia

NA

Yes

Yes

Yes

PK enzyme level, genetic study

Blood transfusion, exchange transfusion

died of sepsis at 3 months old

[Question 8]: Treatment and Management: transfusion regimen not specified (volume, frequency, targets); chelation therapy started late (18 months for deferoxamine), so justify chelation timing based on ferritin trends. Why was liver transplant not considered? Discuss transplant consideration/decision-making and add a brief section on emerging therapies and potential future applications

[Response 8]:

Thank you for the thoughtful comment. Liver transplantation for pyruvate kinase deficiency is considered only after conventional treatments, such as transfusion therapy and splenectomy, have failed. This procedure is reserved for severe cases involving refractory hemolytic anemia, transfusion-dependent iron overload, or serious liver complications that persist despite splenectomy. It is a rare, last-resort option owing to the risks associated with surgery and the need for lifelong immunosuppression (Page 4, Paragraph 1, Lines 130–135).

We hope that the revised manuscript is suitable for publication in Children.

Round 2

Reviewer 1 Report

Comments and Suggestions for Authors

Thank you for honest and prompt answers.

Reviewer 2 Report

Comments and Suggestions for Authors

The revised manuscript is now ready for publication in the current form.